

# The evaluation of factors affecting antibody response after administration of the BNT162b2 vaccine: a prospective study in Japan

Toshiya Mitsunaga[1,2], Yuhei Ohtaki[2], Yutaka Seki[1,3,4], Masakata Yoshioka[4], Hiroshi Mori[5], Midori Suzuka[6], Syunsuke Mashiko[2], Satoshi Takeda[2] and Kunihiro Mashiko[1]

[1] Department of Emergency Medicine, Association of EISEIKAI Medical and Healthcare Corporation Minamitama Hospital, Tokyo, Japan
[2] Department of Emergency Medicine, Jikei University School of Medicine, Tokyo, Japan
[3] Department of Cardiology, Association of EISEIKAI Medical and Healthcare Corporation Minamitama Hospital, Tokyo, Japan
[4] Department of Internal Medicine, Association of EISEIKAI Medical and Healthcare Corporation Minamitama Hospital, Tokyo, Japan
[5] Department of Medical Engineering, Association of EISEIKAI Medical and Healthcare Corporation Minamitama Hospital, Tokyo, Japan
[6] Department of Pharmacy, Association of EISEIKAI Medical and Healthcare Corporation Minamitama Hospital, Tokyo, Japan

Corresponding author
Toshiya Mitsunaga,
toshiya.m@jikei.ac.jp

## ABSTRACT

The aim of this study was to evaluate the antibody reaction after administration of the BNT162b2 vaccine, and to reveal the factors that affect antibody production. This prospective study was carried out in the Association of EISEIKAI Medical and Healthcare Corporation Minamitama Hospital, in Tokyo, Japan, from April 15, 2021 to June 09, 2021. All our hospital's workers who were administered the BNT162b2 vaccine as part of a routine program were included in this study. We calculated the anti-SARS-CoV-2 spike-specific antibody titter (1) before vaccination, (2) 7 to 20 days after the first vaccination, and (3) A total of 7 to 20 days after the second vaccination. The low-antibody titer group (LABG) was defined as the group having less than 25 percentiles of antibody titer. Univariate and Multivariate logistic regression analysis were performed to ascertain the effects of factors on the likelihood of LABG. A total of 374 participants were eventually included in our study, and they were divided into 94 LABG and 280 non-LABG. All samples showed significant antibody elevation in the second antibody test, with a mean value of 3,476 U/mL. When comparing the LABG and non-LABG groups, the median age, blood sugar, and HbA1c were significantly higher in the LABG group. The rates of participants with low BMI (<18.5) and high BMI (>30) were significantly higher in the LABG group. The proportion of chronic lung disease, hypertension, diabetes, dyslipidemia, autoimmune disease, and cancer were significantly higher in the LABG group. Although there was no significant difference confirmed with respect to the exercise hours per day, the proportion of participants that did not perform outdoor exercises was significantly higher in the LABG group. The time interval between the second vaccination and the second antibody test, and between the first and the second vaccination was significantly longer in the non-LABG group. In the multivariate

logistic regression analysis, older than 60 years, the past history of hypertension, HbA1c higher than 6.5%, and lack of outdoor exercises were significant suppressors of antibody responses, whereas the length of days from the first to the second vaccination longer than 25 days promoted a significant antibody response. Again, our single-center study demonstrates that older than 60 years, hypertension, HbA1c higher than 6.5%, and lack of outdoor exercises were significant suppressors of antibody responses, whereas the length of days from the first to the second vaccination longer than 25 days promoted a significant antibody response. Evidence from multi-center studies is needed to develop further vaccination strategies.

## INTRODUCTION

The severe acute respiratory syndrome coronavirus 2 (SARS-CoV-2), which was first detected in Wuhan, China, has spread all over the world significantly faster. By the end of April 2021, the number of cases infected with SARS-CoV-2 had exceeded 150 million, with more than 3.1 million deaths (mortality rate: 2.1%) (*World Health Organization, 2021*). According to the Ministry of Health, Labour and Welfare of Japan, the number of coronavirus 2019 (COVID-19) cases confirmed by 30 April 2021 in Japan was 588,900, with 10,226 deaths (mortality rate: 1.74%) (*The Ministry of Health, Labour & Welfare, Japan, 2021*). Many developed countries in cooperation with the World Health Organization (WHO) have been trying to reduce the number of COVID-19 cases. However, because of its strong infectivity, it has been very challenging to control this disease.

The majority of patients infected with COVID-19 were mild cases, but approximately 5% of cases progressed to severe conditions, and approximately 2% died (*Li et al., 2020*). Several studies have suggested that the risk factors for severe COVID-19 involve individual's age, gender (males are at higher risk compared to females), smoking, and pre-existing conditions such as obesity, diabetes, chronic lung disease, hypertension, dyslipidemia, and chronic kidney disease (*Matsunaga et al., 2020*; *Mi et al., 2020*; *Liang et al., 2020*; *Lippi & Henry, 2020*; *Myers et al., 2020*; *Fadini et al., 2020*; *Zheng et al., 2020*; *Popkin et al., 2020*). Although several treatment strategies, including steroids and antiviral drugs, have been developed, vaccination is still the most important means of preventing COVID-19 infection and aggravation in people with these risk factors. BNT162b2 is a new generation vaccine with nucleoside-modified RNA molecules encoding the full-length SARS-CoV-2 spike glycoprotein (*Polack et al., 2020*). A previous study showed that the efficacy of this vaccine in preventing COVID-19 is approximately 95%, and thus extremely high (*Polack et al., 2020*). Typically, people have to be vaccinated twice to boost their immunity, and the study carried by *Walsh et al. (2020)* showed that spike binding IgG titer increased dramatically and reached a plateau after only 7 days from the second dose. However, the major studies of COVID-19 vaccines are carried out in non-Asian

countries, and the true efficacy of this vaccine in Asian people is still unclear (*Polack et al., 2020*; *Walsh et al., 2020*).

Certain studies have been identifying the factors that promote or suppress antibody response post vaccination include obesity, age, sleep duration etc. A study performed by *Pellini et al. (2021)* showed that age and gender, which were the risk factors for severe COVID-19, may also be interfering factors for SARS-CoV-2 vaccine immunogenicity. However, the population of this study was relatively small and only included non-Asian populations. In contrast, the studies performed by *Tanja et al. (2003)* and *Prather et al. (2021)* demonstrated that sleep duration and sleep quality, which reflect the circadian rhythm, were positively correlated to antibody production after Influenza vaccination or Hepatitis A vaccination, respectively. However, to our knowledge, there have been no studies that evaluate the relationship between the factors involved in the daily life rhythm and the antibody response after COVID-19 vaccination.

Therefore, the aim of this study was to evaluate the anti-SARS-CoV-2 spike-specific antibody reaction following the administration of BNT162b2 vaccine, and to reveal the factors that affect antibody response in a Japanese population.

## MATERIALS & METHODS

### Study design

This prospective study was carried out between April 15, 2021 and June 9, 2021 at the Association of EISEIKAI Medical and Healthcare Corporation Minamitama Hospital, a secondary emergency medical institution. The protocol for this research project was approved by a suitably constituted Ethics Committee of the institution and conforms to the provision of the Declaration of Helsinki (Committee of Association of EISEIKAI Medical and Healthcare Corporation Minamitama Hospital, Approval No. 2020-Ack-19), and written consent was obtained from all the human subjects.

### Study setting and population

All our hospital's workers, who were administered with the BNT162b2 vaccine (COMIRNATY® (Tozinameran)) as part of a routine program, were included in this study. The exclusion criteria were as follows: (1) cases that we could not obtained informed consent, (2) cases with past COVID-19 infection, (3) cases whose antibody titer before vaccination was elevated, (4) cases with new COVID-19 infection after vaccination, and (5) cases who could not provide a blood sample within the expected deadline (*i.e.*, 20 days post-vaccination).

### Data sources and measurements

A five-ml blood sample was drawn from the intermediate cubital vein, and we determined the anti-SARS-CoV-2 antibody titer (Elecsys® Anti-SARS-CoV-2 S RUO, Roche Diagnostics K.K., Basel, Switzerland) (a) before vaccination (baseline), (b) A total of 7 to 20 days after the first vaccination (first antibody test), and (c) A total of 7 to 20 days after the second vaccination (second antibody test). We performed biochemical

examinations (AST, aspartate aminotransferase; ALT, alanine aminotransferase; γ-GT, γ-glutamyl transpeptidase; Alb, albumin; TG, triglyceride; HDL-C, high density lipoprotein cholesterol; LDL-C, low density lipoprotein cholesterol; Cr, Creatinine; BS, Blood Sugar; HbA1c, Hemoglobin A1c; CRP, C-reactive protein) and we also calculated the blood cell count (White Blood Cells, Hemoglobin, Hematocrit, and Platelets) before vaccination.

Self-reports were used to register participants' information with respect to their age, gender, past medical histories (1. Chronic lung disease (Chronic obstructive pulmonary disease: COPD), 2. Chronic lung disease (non-COPD), 3. Cardiac disease, 4. Hypertension, 5. Diabetes, 6. Dyslipidemia, 7. Liver disease (hepatitis, cirrhosis, and any liver disorder), 8. Chronic kidney disease: CKD, 9. Autoimmune disease, 10. Cancer), medication (1. Antihypertensive drug, 2. Antidiabetic drug, 3. Antilipid drug, 4. Antiplatelet and Anticoagulant drug, 5. Immunosuppressive drug, 6. Immunoglobulin preparations), smoking habits (current smoking history: smoking more than once a week within a year, past smoking history: smoking more than once a week over a year ago), habits of drinking alcohol (drinking more than once a week), sleep duration per day, the quality of sleep (good or disturbed), exercise (both of indoor and outdoor exercises) hours per day, the number of outdoor exercises (all outdoor exercises such as camping, mountaineering, scuba diving, skiing, or ball games etc.) days per week, and the length of days for vaccination and antibody test. Body mass index (BMI) was calculated based on the height and body weight which were actually measured before vaccination.

The diagnostic definition of the above diseases conformed to the WHO guideline, and we defined obesity as follows: (1) Underweight: BMI < 18.5, (2) Normal range: $18.5 \leq$ BMI < 25, (3) Pre-obese: $25 \leq$ BMI < 30, (4) Obese class I: $30 \leq$ BMI < 35, (5) Obese class II: $35 \leq$ BMI < 40, (6) Obese class III: $40 \leq$ BMI (*World Health Organization, 2020*).

The low-antibody titer group (LABG) was defined as the group having less than 25 percentiles of antibody titer, whereas non-LABG was defined as having more than 25 percentiles of antibody titer.

## Statistical analysis

A sample size of 460 participants was determined based upon 70% power, 0.05 significance level, 0.3 effect size, three allocation ration, and 20% attrition. Unadjusted analysis evaluated between low and non-low-antibody titer group using the Student's $t$ test and Mann–Whitney $U$ test for continuous variables, which were described as medians and interquartile ranges (IQR), and Fisher's exact test or Pearson's $\chi^2$ test for categorical variables, which were described as numbers and percentages. Univariate and Multivariate logistic regression analysis was performed to ascertain the effects of factors on the likelihood of LABG. Odds ratios and corresponding 95% confidence intervals were calculated. A $p$ value of less than 0.05 was considered to indicate statistical significance. Data were analyzed with the Statistical Package for the Social Sciences, version 26.0 (SPSS, Chicago, IL, USA).
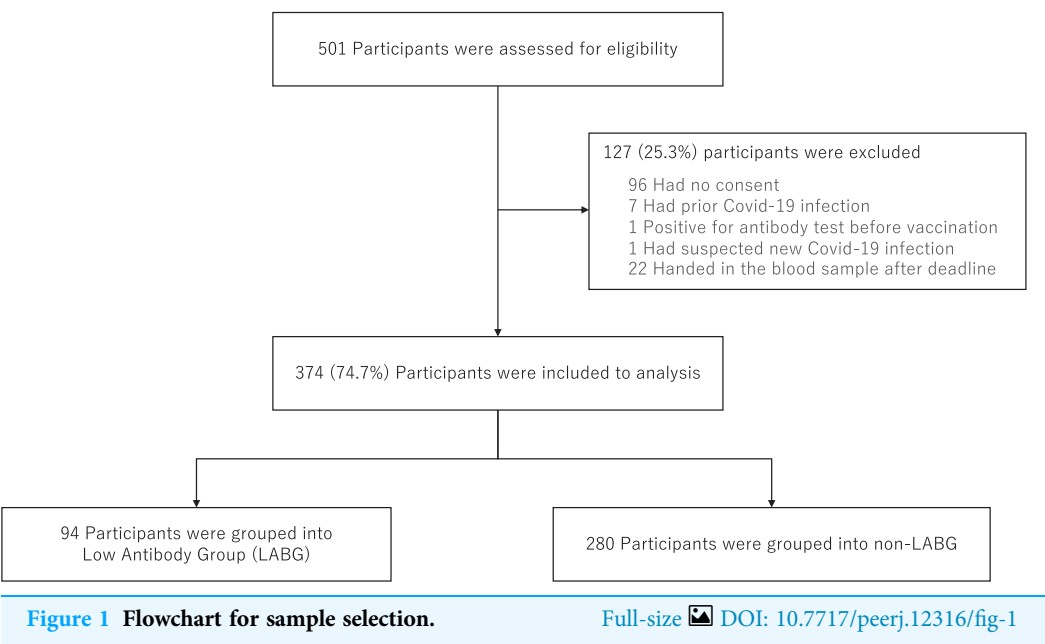

**Figure 1  Flowchart for sample selection.**

## RESULTS

Of the 501 participants who fulfilled the inclusion criteria of this study, we excluded 96 participants because we did not obtain consents, seven participants because of past COVID-19 infection, one participant because of antibody titer elevation prior to the vaccination, one participant because of suspected new COVID-19 infection after first vaccination with unexplained abnormal antibody elevation, and 22 participants because they could not provide a blood sample within the respective deadline. Finally, 374 participants were analyzed that were divided into 94 LABG and 280 non-LABG (Fig. 1). In addition, we were unable to obtain the information on adverse reactions in 15 of 374 participants.

The baseline characteristics are shown in Table 1. The median age (interquartile range) of the participants was 36 (16.0) years, and 110 (29.4%) participants were males. Furthermore, 53 (14.2%) participants with pre-obesity (BMI: 25–30) and 19 (5.0%) participants with obesity (BMI: >30) and zero participants belonging in the obese class III were identified. With respect to comorbidities, 118 (31.6%) participants had past medical histories, chronic lung disease (35 cases: 9.4%), hypertension (30 cases: 8.0%), and dyslipidemia (16 cases: 4.3%). The number of systemic adverse reactions of second vaccination was higher than that of first vaccination. The median duration (interquartile range) between the time from the first vaccination to the first antibody test, and from the second vaccination to the second antibody test was 8 (1) days. Moreover, the median length of days from the first to the second vaccination was 22 (3) days.

Almost all antibody titer was not elevated in the first antibody test, and only 23 (6.2%) samples exhibited a slight positive antibody response, with a mean value of 0.41 U/mL. All samples showed significant antibody elevation in the second antibody test, with a mean value of 3,476 U/mL (Fig. 2).

**Table 1 Baseline characteristics of the study population.**

| | Total population ($n$ = 374) median (interquartile range) |
|---|---|
| Age, years | 36 (16.0) |
| Sex ($n$ (%)) | |
| Male | 110 (29.4) |
| Female | 264 (70.6) |
| Body Mass Index (BMI), kg/m$^2$ | |
| BMI < 18.5 | 34 (9.1) |
| 18.5 ≦ BMI < 25 | 268 (71.7) |
| 25 ≦ BMI < 30 | 53 (14.2) |
| 30 ≦ BMI < 35 | 14 (3.7) |
| 35 ≦ BMI < 40 | 5 (1.3) |
| Past medical histories ($n$ (%)) | |
| Chronic lung disease (COPD) | 1 (0.3) |
| Chronic lung disease (non-COPD) | 34 (9.1) |
| Cardiac disease | 6 (1.6) |
| Hypertension | 30 (8.0) |
| Diabetes | 6 (1.6) |
| Dyslipidemia | 16 (4.3) |
| Liver disease | 6 (1.6) |
| Chronic kidney disease | 4 (1.1) |
| Autoimmune disease | 7 (1.9) |
| Cancer | 8 (2.1) |
| Medication ($n$ (%)) | |
| Antihypertensive drug | 28 (7.5) |
| Antidiabetic drug | 5 (1.3) |
| Antilipid drug | 13 (3.5) |
| Anticoagulant/Antiplatelet drug | 2 (0.5) |
| Immunosuppresive drug | 3 (0.8) |
| Immunoglobulin preparations | 4 (1.1) |
| Current smoking ($n$ (%)) | 70 (18.7) |
| Past smoking ($n$ (%)) | 64 (17.1) |
| Drinking alcohol ($n$ (%)) | 229 (61.2) |
| Laboratory test | |
| WBC (×10$^3$/μL) | 6.35 (2.0) |
| Hb (g/dL) | 13.9 (1.7) |
| Hct (%) | 40.2 (4.5) |
| PLT (×10$^4$/μL) | 25.2 (7.3) |
| Alb (g/dL) | 4.6 (0.4) |
| AST (U/L) | 19.0 (6.0) |
| ALT (U/L) | 15.0 (12.0) |
| γ-GT (U/L) | 17.0 (12.8) |
| Cr (mg/dL) | 0.6 (0.2) |

| Table 1 (continued) | |
|---|---|
| | **Total population (n = 374) median (interquartile range)** |
| CRP (mg/dL) | 0.03 (0.05) |
| BS (mg/dL) | 90 (13.0) |
| HbA1c (%) | 5.5 (0.4) |
| HDL-C (mg/dL) | 65.0 (19.0) |
| LDL-C (mg/dL) | 107 (35.0) |
| TG (mg/dL) | 74.0 (59.0) |
| Adverse reactions (n (%)) | |
| After 1st dose | |
| Local | 352 (98.1) |
| Systemic | 215 (59.9) |
| After 2nd dose | |
| Local | 351 (97.8) |
| Systemic | 324 (90.3) |
| Sleep duration (SD) (n (%)) | |
| SD $\leq$ 2 h | 1 (0.3) |
| 3 h $\leq$ SD < 4 h | 64 (17.1) |
| 5 h $\leq$ SD < 7 h | 297 (79.4) |
| 8 h $\leq$ SD < 10 h | 12 (3.2) |
| The quality of sleep (n (%)) | |
| Good | 232 (62.0) |
| Disturbed | 142 (38.0) |
| The exercise hours (EH) per day (n (%)) | |
| EH < 30 min | 271 (72.5) |
| 30 min $\leq$ EH < 1 h | 79 (21.1) |
| 1 h $\leq$ EH | 24 (6.4) |
| The numbers of outdoor exercises days per week (n (%)) | |
| No | 199 (53.2) |
| More than once | 175 (46.8) |
| The length of days for vaccination and antibody test, days | |
| 1st vaccination to 1st antibody test | 8 (1) |
| 2nd vaccination to 2nd antibody test | 8 (1) |
| 1st vaccination to 2nd vaccination | 22 (3) |

**Note:**

Data are presented as the median (interquartile range) for continuous variables and the number (%) for categorical variables. COPD, Chronic obstructive pulmonary disease; WBC, White blood cells; Hb, Hemoglobin; Hct, Hematocrit; PLT, Platelets; Alb, Albumin; AST, Asparate aminotransferase; ALT, alanine aminotransferase; γ-GT, γ-glutamyl transpeptidase; Cr, Creatinine; CRP, C-reactive protein; BS, Blood sugar; HbA1c, Hemoglobin A1c; HDL-C, High density lipoprotein cholesterol; LDL-C, Low density lipoprotein cholesterol; TG, Triglyceride.

Figure 3 shows the distribution of log antibody titers post second vaccination by relevant age groups. By the age of 50s, the median of the log antibody titers decreases slowly and significantly. In the group over 60 years, a further decrease in the log antibody titer was observed.

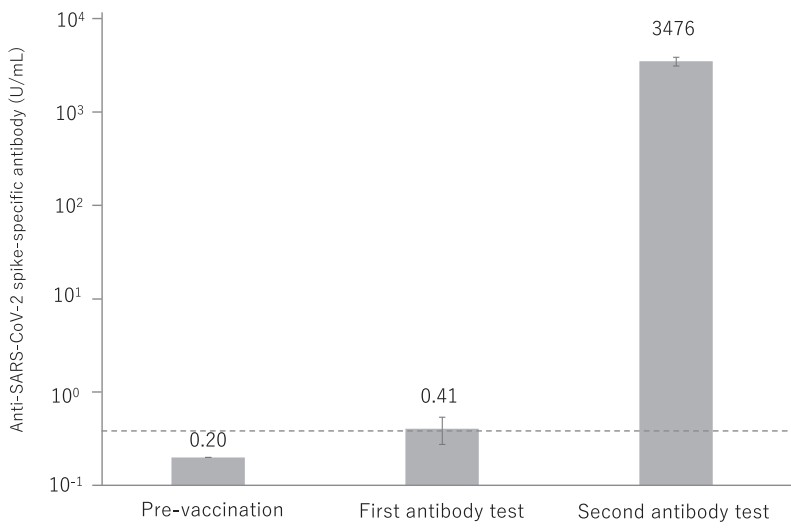

**Figure 2 Anti-SARS-CoV-2 spike-specific antibody response to BNT162b2 vaccination.** A total of 374 participants were administered the BNT162b2 vaccine. Serum samples were obtained before injection and 7 to 20 days after the first and second vaccination. Each bar shows the geometric mean concentrations of anti-SARS-CoV-2 spike-specific antibody (lower limit of quantitation, 0.40; dashed line). The top of the vertical bar represents the mean with a 95% confidence interval (I bar). For values below the lower limit of quantification (LLOQ) = 0.40, LLOQ/2 were included in the calculation.

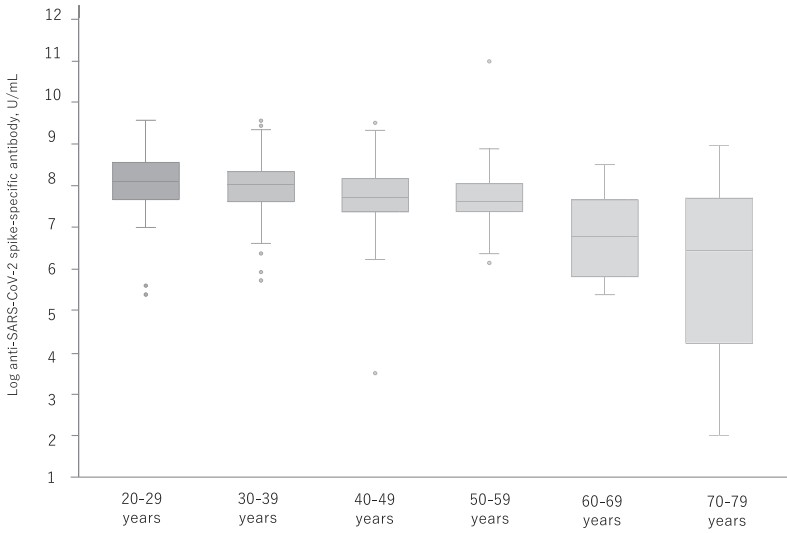

**Figure 3 The distribution of antibody titers post second administration of BNT162b2 vaccine by relevant age groups.** $p < 0.05$: 20–29y *vs.* 50–59y, 30–39y *vs.* 40–49y. $p < 0.01$: 20–29y *vs.* 40–49y, 20–29y *vs.* 60–69y, 20–29y *vs.* 70–79y, 30–39y *vs.* 60–69y, 30–39y *vs.* 70–79y, 40–49y *vs.* 60–69y, 40–49y *vs.* 70–79y, 50–59y *vs.* 60–69y, 50–59y *vs.* 70–79y.

Table 2 shows the comparison between LABG and non-LABG. The median age and HbA1c were significantly higher in the LABG group ($p < 0.001, < 0.01$). The proportion of males (62 (33.2%) *vs.* females (48 (25.7%)) was greater in the LABG group, but with no significant differences ($p = 0.38$). Moreover, the proportion of participants with low BMI (<18.5) and high BMI (>30) were significantly higher in the LABG group ($p < 0.05$).

**Table 2 Comparison of parameters between LABG and non-LABG.**

| | Median (interquartile range) | | |
| --- | --- | --- | --- |
| | LABG (*n* = 94) | non-LABG (*n* = 280) | *p* value |
| Age, years | 41.5 (16.0) | 35 (15.0) | <0.001 |
| Sex (*n* (%)) | | | 0.38 |
| Male | 31 (33.0) | 79 (28.2) | |
| Female | 63 (67.0) | 201 (71.8) | |
| Body Mass Index: BMI (*n* (%)) | | | |
| BMI < 18.5 | 12 (12.8) | 22 (7.9) | <0.05 |
| 18.5 ≦ BMI < 25 | 61 (64.9) | 207 (73.9) | |
| 25 ≦ BMI < 30 | 14 (14.9) | 39 (13.9) | |
| 30 ≦ BMI < 35 | 3 (3.2) | 11 (3.9) | |
| 35 ≦ BMI < 40 | 4 (4.3) | 1 (0.4) | |
| Past medical histories (*n* (%)) | | | |
| Chronic lung disease (COPD) | 1 (1.1) | 0 (0) | <0.05 |
| Chronic lung disease (non-COPD) | 11 (11.7) | 23 (8.2) | |
| Cardiac disease | 0 (0.0) | 6 (2.1) | |
| Hypertension | 15 (16.0) | 15 (5.4) | |
| Diabetes | 5 (5.3) | 1 (0.4) | |
| Dyslipidemia | 9 (9.6) | 7 (2.5) | |
| Liver disease | 2 (2.1) | 4 (1.4) | |
| Chronic kidney disease | 0 (0.0) | 4 (1.4) | |
| Autoimmune Disease | 4 (4.3) | 3 (1.1) | |
| Cancer | 3 (3.2) | 5 (1.8) | |
| Medication (*n* (%)) | | | 0.08 |
| Antihypertensive drug | 13 (13.8) | 15 (5.4) | |
| Antidiabetic drug | 5 (5.3) | 0 (0.0) | |
| Antilipid drug | 8 (8.5) | 5 (1.8) | |
| Anticoagulant/Antiplatelet drug | 1 (1.1) | 1 (0.4) | |
| Immunosuppresive drug | 1 (1.1) | 2 (0.7) | |
| Immunoglobulin preparations | 4 (4.3) | 0 (0.0) | |
| Smoking (*n* (%)) | | | 0.74 |
| Current smoking (*n* (%)) | 20 (21.3) | 50 (17.9) | |
| Past smoking (*n* (%)) | 20 (21.3) | 44 (15.7) | |
| Drinking Alcohol (*n* (%)) | 62 (66.0) | 167 (59.6) | 0.28 |
| Laboratory test | | | |
| WBC (×10$^3$/μL) | 6.4 (2.4) | 6.3 (1.9) | 0.94 |
| Hb (g/dL) | 14.1 (1.4) | 13.8 (1.9) | 0.37 |
| Hct (%) | 40.9 (4.5) | 40.0 (4.3) | 0.44 |
| PLT (×10$^4$/μL) | 25.2 (6.8) | 25.2 (7.3) | 0.99 |
| Alb (g/dL) | 4.6 (0.4) | 4.6 (0.4) | 0.06 |
| AST (U/L) | 19.0 (7.5) | 19.0 (5.0) | 0.20 |
| ALT (U/L) | 17.0 (11.8) | 15.0 (11.0) | 0.05 |

(*Continued*)

| | Median (interquartile range) | | |
| --- | --- | --- | --- |
| | LABG ($n = 94$) | non-LABG ($n = 280$) | *p* value |
| γ-GT (U/L) | 19.0 (16.0) | 16.0 (11.3) | <0.001 |
| Cr (mg/dL) | 0.66 (0.2) | 0.63 (0.2) | 0.67 |
| CRP (mg/dL) | 0.03 (0.04) | 0.03 (0.05) | 0.29 |
| BS (mg/dL) | 91.0 (17.5) | 90.0 (12.0) | 0.05 |
| HbA1c (%) | 5.6 (0.5) | 5.5 (0.4) | <0.01 |
| HDL-C (mg/dL) | 65.0 (20.0) | 65.0 (19.0) | 0.90 |
| LDL-C (mg/dL) | 107.0 (33.8) | 107.0 (36.0) | 0.80 |
| TG (mg/dL) | 75.0 (65.5) | 73.0 (54.0) | 0.67 |
| Adverse reactions (*n* (%)) | | | 0.71 |
| After 1st dose | | | |
| Local | 86 (96.6) | 266 (98.5) | |
| Systemic | 45 (50.6) | 170 (63.0) | |
| After 2nd dose | | | |
| Local | 88 (98.9) | 263 (97.4) | |
| Systemic | 78 (87.6) | 246 (91.1) | |
| Sleep duration (SD) (*n* (%)) | | | |
| SD ≦ 2 h | 1 (1.1) | 0 (0) | 0.33 |
| 3 h ≦ SD < 4 h | 16 (17.0) | 48 (17.1) | |
| 5 h ≦ SD < 7 h | 75 (79.8) | 222 (79.3) | |
| 8 h ≦ SD < 10 h | 2 (2.1) | 10 (3.5) | |
| The quality of sleep (*n* (%)) | | | 0.75 |
| Good | 57 (60.6) | 175 (62.5) | |
| Disturbed | 37 (39.4) | 105 (37.5) | |
| The exercise hours (EH) per day (*n* (%)) | | | |
| EH < 30 min | 66 (70.2) | 205 (73.2) | 0.82 |
| 30 min ≦ EH < 1 h | 22 (23.4) | 57 (20.4) | |
| 1 h ≦ EH | 6 (6.4) | 18 (6.4) | |
| The numbers of outdoor exercises days per week (*n* (%)) | | | <0.001 |
| No | 64 (68.1) | 135 (48.2) | |
| More than once | 30 (31.9) | 145 (51.8) | |
| The length of days from vaccination to antibody test, days | | | |
| 1st vaccination to 1st antibody test | 8 (1) | 8 (1) | 0.90 |
| 2nd vaccination to 2nd antibody test | 8 (0) | 8(1) | <0.05 |
| 1st vaccination to 2nd vaccination | 22 (2) | 22 (3) | <0.05 |

Note:
Data are presented as the median (interquartile range) for continuous variables and the number (%) for categorical variables. COPD, Chronic obstructive pulmonary disease; WBC, White blood cells; Hb, Hemoglobin; Hct, Hematocrit; PLT, Platelets; Alb, Albumin; AST, Asparate aminotransferase; ALT, alanine aminotransferase; γ-GT, γ-glutamyl transpeptidase; Cr, Creatinine; CRP, C-reactive protein; BS, Blood sugar; HbA1c, Hemoglobin A1c; HDL-C, High density lipoprotein cholesterol; LDL-C, Low density lipoprotein cholesterol; TG, Triglyceride.

**Table 3 Univariate and multivariate logistic regression analysis of factors associated with low-antibody titer after vaccination.**

| Predictor | Univariate | | Multivariate | |
|---|---|---|---|---|
| | Odds ratio (95% CI) | *p* value | Odds ratio (95% CI) | *p* value |
| Age ≧ 60 years (*vs.* < 60 years) | 4.70 [1.30–17.05] | <0.05 | 4.99 [1.25–19.85] | <0.05 |
| Sex | | | | |
| Male (*vs.* Female) | 1.25 [0.76–2.07] | 0.38 | | |
| Body Mass Index: BMI | | | | |
| BMI ≧ 30 (*vs.* < 30) | 1.80 [0.69–4.70] | 0.23 | | |
| Past Medical Histories | | | | |
| Hypertension (*vs.* no Hypertension) | 3.35 [1.57–7.16] | <0.01 | 2.58 [1.11–6.02] | <0.05 |
| Current smoking (*vs.* no Current smoking) | 1.24 [0.70–2.22] | 0.46 | | |
| Laboratory test | | | | |
| HbA1c ≧ 6.5% (*vs.* < 6.5) | 14.72 [3.12–69.43] | <0.001 | 10.99 [2.07–58.20] | <0.01 |
| Sleep duration (SD) | | | | |
| SD < 4 h (*vs.* ≧ 4 h) | 1.07 [0.58–1.96] | 0.83 | | |
| The exercise hours (EH) per day | | | | |
| EH < 30 min (*vs.* ≧ 30 min) | 0.86 [0.51–1.44] | 0.57 | | |
| The numbers of outdoor exercises days per week | | | | |
| No (*vs.* more than once per week) | 2.29 [1.40–3.75] | <0.001 | 2.39 [1.41–4.03] | <0.01 |
| The length of days from vaccination to antibody test, days | | | | |
| 1st vaccination to 2nd vaccination > 25 days (*vs.* ≦ 25 days) | 0.49 [0.27–0.88] | <0.05 | 0.47 [0.25–0.89] | <0.05 |

**Note:**
COPD, Chronic obstructive pulmonary disease; WBC, White blood cells; Hb, Hemoglobin; Hct, Hematocrit; PLT, Platelets; Alb, Albumin; AST, Asparate aminotransferase; ALT, alanine aminotransferase; γ-GT, γ-glutamyl transpeptidase; Cr, Creatinine; CRP, C-reactive protein; BS, Blood sugar; HbA1c, Hemoglobin A1c; HDL-C, High density lipoprotein cholesterol; LDL-C, Low density lipoprotein cholesterol; TG, Triglyceride.

The proportion of chronic lung disease, hypertension, diabetes, dyslipidemia, autoimmune disease, and cancer was significantly higher in the LABG group ($p < 0.05$). There was no significant difference in the proportion of adverse reactions between the two groups. Although no significant difference was confirmed in the hours of exercise per day, the proportion of participants with no-outdoor exercises was significantly higher in the LABG group ($p < 0.001$). Finally, the length of days from the second vaccination to the second antibody test, and from the first to the second vaccination was significantly longer in the non-LABG group ($p < 0.05$, $< 0.05$).

Table 3 shows the univariate and multivariate logistic regression analysis of the factors associated with LABG. In the univariate logistic regression analysis, older than 60 years, the past history of hypertension, HbA1c higher than 6.5%, and lack of outdoor exercises were a significant suppressor of the antibody response, and their odds ratios were 4.70, 3.35, 14.72, and 2.29. On the other hand, the length of days from the first to the second vaccination longer than 25 days promoted a significant antibody response, and the odds ratio was 0.49. In addition, the multivariate logistic regression analysis, showed that older than 60 years, HbA1c higher than 6.5% and lack of outdoor exercises were significant suppressors of the antibody response, and their odds ratios were 4.99, 2.58, 10.99, and 2.39. On the other hand, the length of days from the first to the second
vaccination longer than 25 days was promoted a significant antibody reaction, and the odds ratio was 0.47.

## DISCUSSION

Our data showed that the antibody titer was dramatically elevated after the second administration of the BNT162b2 vaccine at thirty micrograms in Japanese people. Moreover, our study identified that older age, past hypertension history, HbA1c more than 6.5%, and performing no-outdoor exercises were significant suppressors of antibody response, as opposed to the duration between the first and the second vaccination longer than 25 days which were found to promote a significant antibody response. This is the first study that evaluated the antibody response in a relatively large cohort following mRNA vaccination for SARS-CoV-2 in Japan, and assessed various factors in terms of promoting or suppressing a subsequent antibody response.

According to previous studies, the mean S1-Binding IgG titers at 21 days post-administration of first vaccination at thirty micrograms was 1,265 U/ml in participants eighteen to 55 years of age, and that in participants 65 to 85 years of age was lower at 329 U/ml. Furthermore, the mean S1-Binding IgG titers at 7 days post-administration of second vaccination at 30 micrograms dramatically elevated at 9,136 U/ml in participants 18 to 55 years of age, and that in participants 65 to 85 years of age was slightly lower at 7,985 U/ml (*Walsh et al., 2020*). In the present study, participants demonstrated that positive anti-SARS-CoV-2 spike-specific antibody reactions rate dramatically increased from 6.2% in the first antibody test to 100% in the second antibody test. The mean anti-SARS-CoV-2 spike-specific antibody titers were 0.2 U/mL in the first antibody test and 3,476 U/mL in the second antibody test, and our findings demonstrate that the second dose of the BNT162b2 two-dose vaccine can also efficiently enhance the immunity of Asian people.

Several studies and guidelines showed that older age (more than 65 years), male, obesity, diabetes, chronic lung disease, hypertension, dyslipidemia, chronic kidney disease and smoking were risk factors for the development of severe COVID-19 (*Matsunaga et al., 2020*; *Mi et al., 2020*; *Liang et al., 2020*; *Lippi & Henry, 2020*; *Myers et al., 2020*; *Fadini et al., 2020*; *Zheng et al., 2020*; *Popkin et al., 2020*).

A study performed by Pellini et al demonstrated that age and male sex may be hampering SARS-CoV-2 vaccine immunogenicity (*Pellini et al., 2021*), but the sample size was not large so that the evidence is limited.

Obesity, which also causes diabetes and hypertension, is one of the most important risk factors of severe COVID-19 by suppressing immune system responses. In this study, we defined obesity in accordance with the WHO guidelines (*World Health Organization, 2020*). A common definition of obesity is important for comparing results of studies performed in different countries, especially when it comes to the identification of immune responses pertaining to racial differences. A previous review by *Milner & Beck (2012)* showed that the excess adipose tissue could block the supply of nutrients to immune cells. Moreover, a study performed by *Vandanmagsar et al. (2011)* revealed that low levels of inflammation and relatively high levels of inflammatory cytokines induced by adipose

tissue can decrease immune responses, and especially T lymphocyte activity (*Vandanmagsar et al., 2011*). These immune cell suppression mechanisms can reduce the subsequent antibody production.

In the present study, obese participants with a BMI over thirty had higher odds ratio for low-antibody response, but was not significant. As we mentioned in the limitation, the reason why there was no significant difference may be related to the decrease in detectability due to the small number of obese people with a BMI of thirty or higher.

Similar to obesity, diabetes is also a significant risk factor element for the development of severe COVID-19, and a report by *Fadini et al. (2020)* showed that the presence of diabetes increases the risk by 2.3 times. In addition, *Berbudi et al. (2020)* showed that hyperglycemia could inhibit IL-6 production, which induces antibodies and T cells, and that this was a mechanism that triggered low-antibody production. Our study showed that the median HbA1c were significantly higher in the LABG group, furthermore, it was the largest factor that suppressed antibody reaction in multivariate regression analysis.

*Gustafson et al. (2020)* exhibited that the immune response toward vaccination is controlled by a delicate balance of effector T cells and follicular T cells, yet the aging process disturbs this balance. Several changes in T cells have been identified that contribute to age-related defects of post-transcriptional regulation, T cell receptor signaling, and metabolic function (*Gustafson et al., 2020*). Similar to Pellini's study, our study revealed that the antibody titer after mRNA SARS-CoV-2 vaccine significantly reduces in particularly participants over 60 years.

The Japanese COVID-19 guideline does not include gender in the risk factors for severe COVID-19, but a report published overseas indicated that male patients are approximately three times more likely to be admitted to intensive care units and have a higher mortality rate (*Mi et al., 2020*).

Females are known to be sensitive to immune responses, including antibody production to infectious diseases, which in turn induces the development of autoimmune diseases (*Fischinger et al., 2019*).

Differences in sex hormones are associated with gender differences in vaccine-induced immunity. For example, testosterone levels and Influenza vaccine antibody titer have been shown to be inversely correlated (*Markle & Fish, 2014*; *Ruggieri et al., 2016*; *Furman et al., 2014*). Genetic differences, as well as sex hormone differences, affect vaccine-induced immunity. The X chromosome expresses 10 times more genes than the Y chromosome, and differences in gene expression between the X and Y chromosomes promote differences in vaccine-induced immunity by gender (*Fischinger et al., 2019*). Our study showed that although not significant the proportion of males was larger in the LABG group, a finding that agrees with current literature.

Furthermore, and although there was no difference in the exercise hours between the two groups, including indoor activities, our study revealed that the proportion of no-outdoor exercises days was significantly larger in the LABG group. Moreover, participants that did not perform outdoor exercises had a significantly higher odds ratio, which was 2.39 in multivariate analysis.

Although there are several differences between exercises, including indoor activities, and outdoor exercises, we focused on exposure to sunlight and the associated vitamin D.

Researchers have long focused on the effect of vitamin D on the activation of the immune system, and a study carried by *Kashi et al. (2021)* showed that there was a positive correlation between plasma vitamin D levels and post-vaccination antibody titer for Hepatitis B. Although no studies have further clarified the relationship between antibody titer and vitamin D levels after vaccination for COVID-19, *Merzon et al. (2020)* found that low plasma vitamin D levels or short duration of exposure to sunlight were associated with an increased risk of COVID-19 infection. Therefore, although vitamin D levels could not be measured in this study, vitamin D levels may be positively correlated with immune responses following administration of the BNT162b2 vaccine.

*Parry et al. (2021)* showed that the antibody response was 3.5-fold higher in cases of delayed second vaccination dose (12 weeks after he first vaccination), and the cellular immune responses were 3.6-fold lower compared to cases with normal vaccination schedule. Our study revealed that a longer interval between the first and the second vaccination had a significant positive correlation with antibody response.

The antibody titer level does not necessarily reflect the immune function against pathogens. However, our study showed that lifestyle improvements such as performing outdoor exercises and reducing complications such as obesity may increase the immune response of the BNT162b2 vaccine, and this fact is especially important for people with risk factors for severe COVID-19. In addition to this, further research of the vaccination schedule is also needed for promoting immune responses after BNT162b2 vaccine administration, especially to older people.

Our study has several limitations. First, we only diagnosed new COVID-19 infections to participants through symptoms of fever and common cold. Therefore, we were unable to identify asymptomatic infections. However, our data showed that the number of antibody-positive cases at baseline, without any previous episode of COVID-19 infection was only one (0.3%), and thus had low statistical power. Second, we only calculated the antibody titer but not the T-cell responses. Consequently, we could not evaluate the effect of these factors to the entire immune system. Third, we calculated the anti-SARS-CoV-2 spike-specific antibody reactions but not the neutralization antibody titer, so that the anti-pathogenic activity has not been evaluated. Fourth, the information may not be accurate as we have collected patient information by self-reports. Finally, we only included relatively healthy medical workers, but did not include participants with severe complications, and this may have reduced the difference between the LABG and non-LABG groups. Further large-scale studies that include participants with several concurrent conditions are needed in the future.

## CONCLUSIONS

Our single-center study demonstrated that older than 60 years, the past history of hypertension, HbA1c higher than 6.5%, and lack of outdoor exercises were significant suppressors of antibody responses, whereas the length of days from the first to the second

vaccination longer than 25 days promoted a significant antibody response. Evidence from multi-center studies is needed to develop further vaccination strategies.

### Funding
The authors received no funding for this work.

### Competing Interests
The authors declare that they have no competing interests.

### Author Contributions
- Toshiya Mitsunaga conceived and designed the experiments, performed the experiments, prepared figures and/or tables, authored or reviewed drafts of the paper, and approved the final draft.
- Yuhei Ohtaki analyzed the data, prepared figures and/or tables, and approved the final draft.
- Yutaka Seki performed the experiments, prepared figures and/or tables, and approved the final draft.
- Masakata Yoshioka analyzed the data, prepared figures and/or tables, and approved the final draft.
- Hiroshi Mori analyzed the data, prepared figures and/or tables, and approved the final draft.
- Midori Suzuka analyzed the data, prepared figures and/or tables, and approved the final draft.
- Syunsuke Mashiko performed the experiments, analyzed the data, authored or reviewed drafts of the paper, and approved the final draft.
- Satoshi Takeda conceived and designed the experiments, authored or reviewed drafts of the paper, and approved the final draft.
- Kunihiro Mashiko conceived and designed the experiments, authored or reviewed drafts of the paper, and approved the final draft.

### Human Ethics
The following information was supplied relating to ethical approvals (*i.e.*, approving body and any reference numbers):

The protocol for this research project was approved by a suitably constituted Ethics Committee of the institution and conforms to the provision of the Declaration of Helsinki (Committee of Association of EISEIKAI Medical and Healthcare Corporation Minamitama Hospital, Approval No. 2020-Ack-19).

### Data Availability
The raw data is available in the Supplementary File.

## Supplemental Information

Supplemental information for this article can be found online at http://dx.doi.org/10.7717/peerj.12316#supplemental-information.

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
