# Peer review of "The evaluation of factors affecting antibody response after administration of the BNT162b2 vaccine: a prospective study in Japan"

_PeerJ, doi:10.7717/peerj.12316_

## Round 0.1 · original submission · Major Revisions

Please pay particular attention to the concerns raised with the statistical analyses.

Reviewer 1 ·

Basic reporting

The authors of the study evaluated antibody responses to The BNT162b vaccine in HCW who received the vaccine in Japan. The data is important as there is limited data amongst different ethnic group, so it will add to this body of literature. However, I have several concerns that I am listing below for your consideration. One major limitation are the models used for the analysis. They need to be carefully considered and represented.

1- In general, I was able to read the paper and understand what was being said, however having said, there are some areas where the sentence construction could be improved upon. For example the background (sentence 100 and 101). I might phrase this as "factors that promote or suppress antibody production post vaccination include obesity, age, sleep duration etc" and then proceed to discuss the studies. Similarly in the discussion, instead of stating a large number of facts it would be important to discuss what the authors want to take away from their paper.
2- Appears to be referenced properly, however, numbering of references would be easier. Perhaps this is no a requirement of the journal
3- Raw data was shared, however, as far as the ethical approval is concerned, I did not note a signature on the English version, perhaps the Japanese version was signed.

Specific points

Under study setting and population
Line 127- Cases with new COVID-19 infection after vaccination- is there a time frame for this.
Line 141- was there any validation done about the self-reported BMI?
Line 143- What is the definition of liver disease
Line 146- smoking history (was pack year captured)
Line 147- What is the definition of alcohol habits (were the number per day/week captured)
Line 148- States quality of meal (was this meant to be sleep)
Line 149- what is the definition of outdoor activity for thsi analysis
I think since a lot of this data is self-reported it should be mentioned as a limitation

Regarding statistical analysis-
This is a major limitation of the paper as I think there is an issue with over-adjustment with multiple variables included in the model. Hence I think the regression analysis should be repeated with a succint set of variables that are non-collinear. For example blood sugar and HbAiC are likely to be collinear and yet they are both adjusted for in this model, would it not be more relevant to use clinically relevant cut-offs (such as <5.7, 5.7 to 6.5, or >6.5, or whatever values are used in Japan to define DM and glucose intolerance)
Some of the variables evaluated in the model have such few numbers it appears that the dataset does not have the power to examine these variables for example cancer, autoimmune disease, liver disease, anticaoagulant therapy, immunosuppressive drugs, immunoglobulin (what is immunoglobulin therapy referring to, convalescent plasma/monoclonal antibody administration against SARS Cov2?).
Similarly for the laboratory tests utilized there appear to be limited rationale to examine for example why hb and hct in the same regression analysis, it appears to me that all of the variables were put into a model and examined. Instead, the authors should critically evaluate what is relevant to their outcome and include these variables.
With 94 participants in the LABG group (it appears that you should be adjusting for only 10 to 12 variables at the most).
I am not sure if the results will hold in a relevant model. These analysis should be re-examined.

In the regression analysis (Reference classes are not mentioned). For example for BMI groups what was the referent group. All the values mentioned have an Unadjusted OR which is not 1.0. Similarly for alcohol what is the referent group, so for example compared to what is the value 1.31 (no alcohol?). All of the referent groups should be clearly outlined in the table


Results-
i- Could the authors provide a description of the distribution of titers post vaccination by relevant age groups, this is an important aspect that seems to be possible with the data collected (perhaps evaluate based on median age distribution)
ii- There is no data on adverse reaction profile of the vaccine- was this captured this seems like a relevant piece of information to include to evaluate if the adverse effect profile of the vaccine is similar among Japanese HCWs

Discussion
Lines 221 to 223- I think the discussion should consider differences between the current study and those cited, rather than merely presenting numbers and concluding that the vaccine responses are similar. For example the Walsh et al article compared differing does of BNT162b vaccine (10, 20. 30 mg), whereas the vaccine under EUA is the 30 microgram dose, which is what I presume was used for the study (which should be mentioned in the paper). In the Walsh et al article, the S1-Binding antibody responses varied according to the age of the participants and also according to the dose of the vaccine. The authors suggest that the median antibody titers were 9136 U/mL after 28 days, but this number applies to those aged 18 to 55 years who received the 30 micrograms of the BNT162b2 vaccine. Also the mentioned level of 0.6 U/mL in the Walsh et al paper was in the 18 to 55 years of age who received the 30 microgram dose and this was before administration of the vaccine, so it is unclear to me how this value can be compared to the value (7 to 21 days after the administration of the vaccine). Further, the assays used in the two studies were different. So, I think these nuances need to be explained in the discussion. I think the important message here is that the antibody responses are better after the second dose of the vaccine, also in my read they appear to be poor after first dose, barely above the detectable level in the figure) suggesting that a strategy of a single dose of vaccine will be unsuccessful in this population. Further the discussion talks about median antibody titers but the figure represents the mean titers. Did they mean to say “mean” titers?

Lines 233- I think this should say “older age” the term age should be qualified. How is the age term used in this paper should be discussed
Line 235- I did a quick pubmed search using the terms “obesity and antibody responses to covid vaccines” and I was able to find articles on this (examples- https://pubmed.ncbi.nlm.nih.gov/34109307/) so I am not sure this statement is entirely accurate or perhaps I am misunderstanding what is being discussed.
Line 249 to 250 – as previously stated I am really unable to understand what some of the reference categories are for the analysis
Line 259 to 261- Why are we discussing triceps thickness when the site of injection is the deltoid
Line 268 to 271- this could be an issue with multiple adjustments too. As above this is an issue that should be rectified.
Line 277- Consider stating as “we age”. What is older for this study, I could not figure this out from the table, is the regression analysis based on every year increase in age or every five or every 10?
Line 304 to 310- I think the authors should clearly state where they hypothesize that they are hypothesizing reasons for their observed association, it appears to me that the authors are hypothesizing that the observations associated with no outdoor activity and lower antibody responses is related to vitamin D levels. Do they have banked serum that they can use to explore this association?
Line 334- By stating CD4 lymphocyte activity, are they meaning T cell responses to vaccine, if so they should correct. Also neutralization antibody levels were not examined which is a limitation of the paper.

Experimental design

As above. I feel the model should be re-run

Validity of the findings

In my opinion, the statistics has some serious issues as discussed above.

Annotated reviews are not available for download in order to protect the identity of reviewers who chose to remain anonymous.

Reviewer 2 ·

Basic reporting

Ok

Experimental design

Sample size is low and no new findings

Validity of the findings

Statistical backup needed

Additional comments

No new finding generated from this data
Only antibody titers wont be sufficient for evaluation of vaccine in population other CMI studies needs to be performed

Annotated reviews are not available for download in order to protect the identity of reviewers who chose to remain anonymous.

---

## Round 0.2 · accepted · Accept

The manuscript was significantly improved by the authors following the reviewers' comments and is now suitable for publication.